# Isolated Severe Neutropenia in Adults, Evaluation of Underlying Causes and Outcomes, Real-World Data Collected over a 5-Year Period in a Tertiary Referral Hospital

**DOI:** 10.3390/medicina60101576

**Published:** 2024-09-26

**Authors:** Linet Njue, Naomi Porret, Annatina Sarah Schnegg-Kaufmann, Luca Francesco Varra, Martin Andres, Alicia Rovó

**Affiliations:** 1Department of Hematology and Central Hematological Laboratory, Inselspital, Bern University Hospital, University of Bern, 3010 Bern, Switzerland; naomiazur.porret@insel.ch (N.P.); annatina.schnegg-kaufmann@insel.ch (A.S.S.-K.); martin.andres@insel.ch (M.A.); alicia.rovo@insel.ch (A.R.); 2IDSC-Insel Data Science Centre, 3010 Bern, Switzerland

**Keywords:** neutropenia, drugs, infections, idiopathic, congenital

## Abstract

*Background and Objectives*: In clinical practice, neutropenia is frequently accompanied by other cytopenia; isolated non-chemotherapy-induced severe neutropenia is less frequent and its differential diagnosis can be challenging. In this real-world study with data collected over a 5-year period in a tertiary referral hospital, we primarily sought to identify underlying causes of isolated severe neutropenia (<0.5 × 10^9^/L). Secondly, we aimed to analyze its management and outcomes. *Materials and Methods*: From 444,926 screened patients, after exclusion of patients with chemotherapy, radiotherapy, hematological neoplasms, additional cytopenia, and benign ethnic neutropenia, we identified and analyzed data from 70 patients (0.015%) with isolated severe neutropenia. We thus confirmed that the occurrence of isolated severe neutropenia is a rare event, even in a tertiary hospital. *Results*: The median age at diagnosis was 34 years (range 1–81) and 65% were female. Acute neutropenia was more frequently observed (n = 46/70, 65.7%); the main underlying causes in this group were drugs (n = 36/46, 78%) followed by infections (n = 10/46, 21.7%). We identified 24 (34.3%) patients with chronic neutropenia. The majority of them (n = 12/24, 50%) had an idiopathic form (CIN), 8/24 (33%) were autoimmune (AIN), and 4/24 (17%) were congenital. *Conclusions*: This study demonstrates the rarity and heterogeneity of isolated severe neutropenia and the steps to consider in its diagnostic work-up and management. Epidemiological characteristics, diagnostic work-up, and management including hospitalizations are described. Due to the high frequency of metamizole-induced neutropenia observed in this study, we want to raise awareness about its use, since this complication generates frequent hospitalizations even in young, otherwise healthy patients. Furthermore, recurrent infections in chronic forms of idiopathic neutropenia were quite common, suggesting a difference in phenotypes and need for therapy consideration depending on the clinical course.

## 1. Introduction

Severe neutropenia, also denominated as agranulocytosis, defined as an absolute neutrophil count (ANC) < 0.5 × 10^9^/L, is of immense clinical relevance due to its association with increased risk of infections in most patients [1,2]. In clinical practice, neutropenia is frequently accompanied by other cytopenia; isolated non-chemotherapy-induced severe neutropenia is less frequent and represents the focus of this research.

Acquired forms of isolated neutropenia are more common than congenital forms, with drugs and toxins being the most common causes in adults [3,4,5]. Post-infectious neutropenia can occur due to a variety of viral infections such as parvovirus, dengue virus, cytomegalovirus (CMV), Epstein-Barr virus (EBV), Human Immunodeficiency Virus (HIV), respiratory syncytial virus (RSV), influenza A and B, severe acute respiratory syndrome coronavirus type 2 (SARS-CoV-2) [6,7,8,9,10,11,12], as well as parasitic [13] and, more rarely, bacterial infections. Dietary causes, as well as autoimmune disorders and chronic idiopathic forms, are other known causes of acquired neutropenia [14].

Severe congenital neutropenia (SNC) is rare and has a prevalence of approximately 1–2 cases per million inhabitants [15]. Mutations in the ELANE gene are the major cause, occurring in 50 to 60% of these patients, but there have been a large number of other genes described [16,17,18,19]. These patients are diagnosed early in childhood and today, thanks to supportive treatment with GCS-F, they survive and are followed in adult clinics, requiring close monitoring due to their increased risk of clonal hematopoietic diseases such as myelodysplastic syndromes (MDS) and leukemia [20,21].

The association between fever and severe neutropenia and the risk for life-threatening infections is well known, but most treatment guidelines refer to patients receiving cytotoxic chemotherapy [22]. Evidence-based recommendation guidelines in regards to the use of G-CSF in other forms of neutropenia are lacking, and the available recommendations are mostly based on expert opinions [23]. Furthermore, the diagnostic work-up in pediatric patients with severe isolated neutropenia has been largely addressed in the literature [24,25] and the approach in adult patients [23] has also been discussed recently, highlighting the need of a diagnostic work-up adapted to the clinical situation.

In this retrospective study, we primarily sought to identify underlying causes that might account for isolated severe neutropenia in adult patients seen at our tertiary hospital over 5 years. Secondly, we aimed to analyze the management and outcomes.

## 2. Materials and Methods

The hospital database was searched to identify adults with isolated severe neutropenia who were seen between 1 January 2015 and 30 September 2020. We included in this study adult patients aged above 18 years with neutropenia < 0.5 × 10^9^/L treated either in the outpatient or inpatient setting of our tertiary hospital. Concomitant cytopenia (hemoglobin levels < 100 g/L or platelet levels < 100,000/µL), neutropenia due to chemotherapy, radiotherapy, or known hematological neoplasia such as MDS, and benign ethnic neutropenia were exclusion criteria. Patients who denied the hospital’s general consent were excluded from this study. To characterize the duration of neutropenia, we classified the neutropenia as acute or chronic, lasting less than 3 or more than 3 months respectively.

Patient demographic data, ICD-10 diagnosis, laboratory data, particularly complete blood count (CBC) and white blood count (WBC), and diagnostic methods to characterize the neutropenia (bone marrow investigations, cytogenetic and molecular investigations) were evaluated. Furthermore, blood and other samples cultures results were analyzed, as well as radiological investigations used to evaluate infections. Infections, use of G-CSF and hospitalization were analyzed. In women of childbearing age, pregnancies, their management regarding neutropenia, and the outcomes of the mother and the child were investigated.

The study was approved by the local ethics committee of the canton of Bern (Project-ID number: 2020-00314) and was conducted in accordance with the Declaration of Helsinki.

## 3. Statistical Analysis

Statistical analyses were performed using R Software version 4.2.2. The quantitative characteristics of the patients are shown as numbers (*n*) and frequencies (percentage %) in the tables. The numerical/quantitative variables are expressed as medians (range). We used the Wilcoxon test for pairwise comparison of continuous variables and the Chi-square test and Fisher’s exact test for comparison of count data. Data management was performed using Microsoft Excel (2016). This study adopts a descriptive approach.

## 4. Results

From 444,926 screened patients during the selected period, 70 (0.015%) patients with isolated severe neutropenia met the inclusion criteria (Figure 1) and were included in the study. The median age of the patients at neutropenia diagnosis was 34 years (range 1–81). Three patients with SCN were diagnosed in childhood, which explains this wide age range. The majority of the patients were women 45 (64%). The ethnicity data of this population showed that 94% (N = 66) were Caucasian, three were of African descent and one patient was of Asian descent.

### 4.1. Causes of Isolated Neutropenia

Acute neutropenia was more frequently observed (*n* = 46/70, 65.7%) than chronic neutropenia. The main underlying causes in this group were drugs (*n* = 36/46, 78%) followed by infections (*n* = 10/46, 21.7%). We identified 24 (34.3%) patients with chronic neutropenia. The majority of them (*n* = 12/24, 50%) had an idiopathic form, chronic idiopathic neutropenia (CIN), 8/24 (33%) were autoimmune (AIN), and 4/24 (17%) were congenital (Figure 2).

### 4.2. Bone Marrow Investigations and Additional Diagnostic Work-Up

From 70 patients, 20 (28%) underwent bone marrow (BM) investigation, including both aspiration and biopsy (Appendix A). This diagnostic step was mainly performed in patients with chronic evolution (*n* = 19/24, 79%), while only 1/46 (2%) acute cases underwent BM investigation. In 9/20 BM, an additional investigation with flow-cytometry was performed, and the main question was focused on lymphoid neoplasms and blasts. No patient in this cohort presented a hematological malignancy or BM infiltration by another disease. Spleen size was investigated by ultrasound in seven patients. All presented with mild splenomegaly (median spleen size 14.4 cm (range 13–15); four of these patients had chronic neutropenia and three had acute. Flow cytometry of peripheral blood was investigated in twenty-three patients, sixteen chronic and seven acute. The most important search was for lymphoid neoplasms, mainly large granular leukemia (LGL), which was not confirmed in any of the investigated cases.

### 4.3. Drug-Induced Neutropenia

Metamizole was the drug most often associated with acute neutropenia in our cohort (*n* = 14/36, 39%), followed by rituximab (*n* = 4/36, 11%). The median duration of drug-induced neutropenia was 4 days (range 1–43 days), with the longest duration observed in a patient with rituximab-induced neutropenia (43 days). The offending drugs and frequencies are listed in Table 1. Regarding the clinical evolution of drug-induced neutropenia patients, in 34/36 patients (94%) neutropenia resolved during the observation period, one patient was lost to follow-up, and one died soon after neutropenia was diagnosed.

Antibiotic treatment due to concomitant fever was administrated in 12/36 cases (36%), and 8/36 patients (22%) received G-CSF.

A separate analysis of patients with metamizole-induced neutropenia showed the patients had a median age of 30 years (range 21–64) with 57% of the patients being female. Neutropenia in this cohort lasted a median of 2.5 days (range 1–8). Of these patients, 29% received G-CSF whilst 86% were treated with antibiotics (Appendix A).

### 4.4. Post-Infectious Neutropenia

The responsible pathogen was identified in 6/10 patients with post-infectious neutropenia (Table 2); a variety of infectious agents were observed without specific prevalence. The identified responsible pathogens were determined by serological testing. Neutropenia resolved in all of these patients. The clinical characteristic was a short duration of neutropenia of a few days, the longest being 7 days in a patient with a rhinovirus infection. In the remaining 4/10 patients, despite the fact that no pathogen was identified, a viral origin was suggested based on clinical and laboratory findings.

### 4.5. Congenital Neutropenia

We identified four patients with congenital neutropenia. The genetic mutations detected in our cohort were ELANE mutations in two patients, a HAX1 mutation in one patient and a WAS mutation in one patient. Three patients have had recurrent infections since early childhood; from these, two have been treated continuously with G-CSF since then, and one continuously since age 26. Despite severe neutropenia, the fourth patient with a WAS mutation has presented a low rate of infections, receiving G-CSF only during infections.

All these cases underwent a BM investigation. In two patients, additional investigation including cytogenetic and next-generation sequencing (NGS) was performed. NGS testing revealed the presence of the DNMT3A mutation with a low mutation burden of 4% in the patient with a WAS mutation. None showed hematological transformation.

### 4.6. Chronic Idiopathic Neutropenia/Idiopathic Cytopenia of Undetermined Significance-Neutropenia (ICUS-N)

CIN/ICUS-N represented 17% of the investigated cohort. No underlying cause was identified. The majority (83%) were women. The clinical courses were heterogeneous. Seven patients had recurrent infections and needed G-CSF therapy either sporadically or continuously.

### 4.7. Chronic Autoimmune Neutropenia

Finally, eight (11%) patients in the cohort had a concomitant autoimmune disorder, suggesting AIN in these cases. The associated identified autoimmune disorders included: colitis ulcerosa, myasthenia gravis, granulomatosis with polyangiitis, undifferentiated connective tissue disease in three patients, rheumatoid polyarthritis, and necrotizing myositis.

Two of these patients had positive anti-neutrophil antibodies (the patient suffering from concomitant rheumatoid polyarthritis and one of the patients with undifferentiated connective tissue disease). Spontaneous remission of neutropenia was, however, not observed in any of the patients.

### 4.8. Pregnancy in Patients with Chronic Neutropenia

There were 16 women with a median age of 28 at diagnosis (range 17–67) with chronic neutropenia. In total four women, three with CIN and one with AIN diagnosis, had a total of eight pregnancies. One patient was undergoing regular treatment with G-CSF due to recurrent infections; this treatment was continued at her regularly dosage of 3 mcg/kg every 2 weeks during pregnancy without complications.

### 4.9. Infections, Use of G-CSF and Hospitalisation

We observed that 34% (*n* = 24/70) of patients had infections that required hospitalization, but none of these patients required intensive care. A total of 18/70 (26%) patients received G-CSF with different modalities (sporadic or continuously); these were mainly patients with chronic neutropenia (Table 3). Patients with congenital forms were those who had the higher requirements for both G-CSF treatment and hospitalizations. Of note is the high hospitalization rate of 47% observed in drug-induced forms, which are patients who were otherwise healthy. In addition, the use of treatment with G-CSF in this group, which was used in 22% of patients, suggests that they were used to shorten the time of neutropenia in a particularly young population.

## 5. Discussion

The occurrence of isolated severe neutropenia is a rare event, even in a tertiary hospital. This study shows that the underlying etiologies are heterogeneous, with acute forms being the most frequently observed, particularly those induced by drugs, followed by infectious causes. Chronic forms, although very unusual, constituted more than a third of the population evaluated, with idiopathic forms leading this group. The hematological interpretation and management of patients presenting with isolated neutropenia can be challenging since neutropenia may be the initial presentation of a hematological disease or another clinical entity. Indeed, the number of differential diagnoses is extremely broad. Additionally, if the patient also presents with a fever and a critical general condition, it represents an emergency and it is imperative to understand the underlying cause of the neutropenia for further management.

According to current guideline recommendations [23,26], the initial step for the investigation of patients with neutropenia is a detailed past medical and family history. For adult patients, drug history and recent infections are important, as well as work-ups for autoimmune and other disorders that may be associated with neutropenia. After initial investigations, a bone marrow aspiration and a core biopsy should be considered in all patients with unexplained persistent neutropenia. Exclusion of hematological neoplasms, especially before administration of G-CSF, is an important step to consider. In line with these recommendations, part of this cohort was extensively evaluated with BM and other additional investigations, mainly in patients with chronic courses, except for one who was investigated when neutropenia was newly diagnosed. In critically ill patients, diagnostic steps might be rushed by the obvious need for decisions based on specific diagnostic information.

The most common causes of isolated neutropenia in adults are drugs and infections [3,27]. The annual incidence of drug-induced neutropenia is approximately 3 to 16 cases per million patients exposed to drugs [28,29,30,31]. Common repeat offenders include metamizole, ticlopidine, sulfasalazine, carbimazole, clozapine, rituximab and beta-lactam antibiotics [3,4,31].

Our results are in line with the current literature since drugs were the most frequent cause in the entire analyzed cohort. Agranulocytosis is a well-known and potentially lethal side effect of metamizole [32,33,34,35] with an estimated incidence rate of 0.46–1.63 per million person-days [36], underreporting not considered. Though detailed information on the exact dosage and duration of metamizole exposure was not available in this study, it has been shown that dose, in addition to duration of use, may be a risk factor for the development of agranulocytosis associated with metamizole [4,36]. Metamizole is a very efficient analgesic drug, but due to its safety profile, it is banned in many countries such as the United States, Saudi Arabia, Australia, Denmark, and India among others [37,38]. Our data underlines the frequency of this problem. It is paramount that prescribing physicians, as well as patients, understand this possible adverse reaction and discuss alternative analgesia where possible. Management guidelines may also recommend some policy of restricted prescribing and stringent monitoring. Accurate reporting of drug-associated neutropenia is also of great importance, especially in regards to implementing preventive measures to reduce their impact [33,39,40]. Additionally, developing risk stratification tools for managing patients who develop neutropenia, taking into account the patient’s age and co-morbidities amongst other factors, may be helpful to decide the setting of hematological monitoring and care.

As was observed in our cohort, drug-induced neutropenia improves mostly spontaneously once the drug is discontinued, although antibiotics and G-CSF treatment may be indicated.

Post-infectious causes accounted for 10% of all isolated neutropenia cases in our cohort and were the second most common causes of acute forms, with viruses being the main confirmed culprits in 50% of the cases. Many infectious pathogens may cause neutropenia, but the most common infectious etiology, like in our study, is viral [6,7,8].

According to current guidelines [23], patients with isolated acquired neutropenia who have had recent infections should undergo viral and bacterial investigations. Our results are in accordance with this recommendation. Serum testing for viral-specific antibodies is generally considered the gold standard for diagnosis, whereas quantification or monitoring of viral load can help inform assessment of risk and strategies for treatment. We therefore recommend antibody studies as the initial diagnostic step.

Since our data was collected before the peak of the COVID-19 pandemic, we did not have COVID-19-induced neutropenia cases in our cohort. Many cases of neutropenia associated with the COVID-19 virus have, however, been reported [9,10,11,12,41]. Viral-related neutropenia may be mild to severe, and is generally of early onset and short duration. Infections may produce neutropenia by a variety of mechanisms including autoantibodies, direct infiltration of hematopoietic cells, and direct suppression of marrow precursors by toxins elaborated by the infectious agent [2]. We reported one case of post-bacterial neutropenia caused by *Haemophilus influenzae*. This not only highlights the importance of serology testing in the diagnostic work-up of neutropenia [42] to aid correct therapy, but also shows the significance of vaccinations as preventative measures for such complications.

We found that 18.5% of the evaluated patients fulfilled the diagnostic criteria of CIN/ICUS-N [43,44,45]. CIN/ICUS-N is defined as chronic isolated neutropenia lasting ≥3 months and not attributable to drugs, infections, or autoimmune or malignant causes. Numerous studies suggest that the pathophysiology of CIN/ICUS-N is attributable to apoptosis of developing neutrophils in the bone marrow mediated through activated T lymphocytes and excessive production of pro-inflammatory and myelosuppressive cytokines such as IFNγ, TNFα, and Fas-ligand [46,47,48,49]. There are, however, no clinical tests yet available to aid in diagnosis. There was a female predominance in our cohort, as has been described in the literature [43]. We observed that 41% of CIN/ICUS-N patients had recurrent infections, suggesting a difference in phenotypes and a need for therapy consideration depending on the clinical course and not on the neutrophil value alone. A benign course, without transformation to myeloid neoplasia, was observed in all cases.

Primary AIN is rare in adults and is usually clinically benign [50]. CIN and AIN are very similar and sometimes overlapping conditions [44]. Secondary AIN may develop, as seen in 11% of patients in our cohort, in association with systemic autoimmune diseases such as rheumatoid arthritis, systemic lupus erythematosus, immune hemolytic anemia, or Felty syndrome [2,51]. The condition is caused by antibody-mediated destruction of neutrophils [50]. Therapy is based on identifying and treating the primary autoimmune disorder; therefore, correct diagnosis in conjunction with all clinical factors is of great importance. The presence of anti-neutrophil antibodies may aid diagnosis [52,53].

The four congenital neutropenia cases were confirmed using genetic testing. Genetic testing should therefore be considered as part of the work-up for patients who are suspected of having a primary neutropenia disorder after acquired causes are excluded [23,54]. These types of diagnoses are generally made in pediatric age; however, there are patients who reach adulthood without genetic confirmation. Internists should be aware of the availability of existing tests to characterize these patients. The most frequent pathogenic defects in SCN, as seen in our patients, are autosomal dominant mutations in ELANE [18] and autosomal recessive mutations in HAX1 [55]. In one female patient with chronic neutropenia, the Wiskott-Aldrich syndrome (WAS) gene mutation in heterozygote form was detected in adulthood. Very rare cases of neutropenia are associated with activating mutations in the gene encoding the WAS protein, resulting in an X-linked form of SCN [56,57,58]. Our patient had a mild clinical course without recurrent infections or need for G-CSF use.

SCN patients have a high risk of invasive bacterial infections and oral infections. G-CSF is the treatment of choice and has dramatically changed both the prognosis and the quality of life for these patients with a reduction in infections [15,20,59]. Lifetime G-CSF treatment is usually needed in SCN, with individual tailoring of the dose, in order to achieve therapeutic target neutrophil counts ≥1.0 × 10^9^/L. For acquired forms of chronic neutropenia, G-CSF therapy should be recommended for patients with absolute neutrophil counts repeatedly less than 0.5 × 10^9^/L with recurrent mouth ulcers, gum disease, and/or recurrent infections.

None of the SCN patients in our cohort had additional high-risk somatic mutations. Nevertheless, regular clinical assessments to monitor treatment course and detect chromosomal abnormalities, as well as somatic bad-prognosis myeloid mutations, are recommended in all SCN patients, as it is a premalignant condition.

The indication of G-CSF is different in severe acquired neutropenia forms, since its indication is limited to cases with recurrent infections or stomatitis [26,44], only on-demand. We observed that in patients with acute neutropenia, with the exception of one patient who died of probable infection, neutropenia resolved over time, regardless of the use of G-CSF. Noteworthy is the fact that though none of the patients with post-infectious neutropenia received G-CSF, complete resolution of the neutropenia was documented in all cases. These results support the fact that treatment decisions should be based on the underlying etiology.

Accordingly, a majority of the patients with chronic neutropenia received G-CSF continuously with good responses.

The recommended dose of G-CSF in chronic neutropenia patients is 1 to 8 mcg/kg per day, with the effective doses varying by diagnosis: idiopathic: median 1.2 mcg/kg/day; cyclic: median 2.4 mcg/kg/day; congenital: median 7.3 mcg/kg/day [26].

Although G-CSF is safe and efficient in long-term treatment of adult patients with chronic neutropenias, possible side effects may include bone pain, arthralgias, myalgias, and headache with onset a few hours after the injections. Less common adverse events include thrombocytopenia, skin rash, injection site reactions, vasculitis, decreased bone density, and osteoporosis, though fractures are uncommon [15,60].

None of our patients with chronic neutropenia received antibiotic prophylaxis. Experts agree on the concept that there is no major advantage in antibiotic prophylaxis in chronic neutropenia other than to treat the infection at its occurrence.

In regards to pregnancy, though data is limited, the use of G-CSF throughout pregnancy, as we have seen in our cohort, has been documented as safe and well-tolerated with no noticeable side effects [61,62].

Our study has several strengths. Some strengths are the huge number of patients examined, the confirmation of isolated neutropenia with laboratory results, the availability of all laboratory data from additional investigations, and the access to clinical data from all of them, making the cohort very consistent. Another strength of the study is the applicability and relevance of the findings to clinical practice, which can lead to more personalised interventions. Another strength of this study is its generalizability due to the large population investigated. To the best of our knowledge, similar data has not been published to date. This study has, however, a number of limitations, mainly related to its retrospective nature and the low number of patients in some subgroups. Some subgroups may be overrepresented due to the characteristics of our hospital as a referral center. We would like to stress that the diagnostic procedures, as well as the decision to treat an individual patient, were made at the discretion of the treating physician. There is also some data lacking, as not all diagnostic tests were performed and, in some patients, the reason for treatment was not always understandable (Figure 3).

## 6. Conclusions

This research highlights the heterogeneous presentation of severe isolated neutropenia, demonstrating that it can have both acute and chronic courses. Despite the severity of neutropenia in some cases, the overall outcome is benign and fatality is nowadays unusual. The real-world data from this study shed light on the clinical course, management, and outcomes of isolated severe neutropenia, contributing information that may contribute to the understanding and management of this condition. Due to the high frequency observed in this study of metamizole-induced neutropenia, a drug banned in several countries, we want to raise awareness about its use, since this complication generates frequent hospitalizations even in young, otherwise healthy patients. Furthermore, the manuscript underscores the limited existing guidelines regarding the use of granulocyte colony-stimulating factor (G-CSF) in non-chemotherapy-induced neutropenia and stresses the need for more comprehensive recommendations and research on the use of G-CSF in various forms of neutropenia. It also provides some information on the safety and tolerability of G-CSF use during pregnancy in women with chronic neutropenia considering the need to report these data due to the lack of available data in the literature.

## Figures and Tables

**Figure 1 medicina-60-01576-f001:**
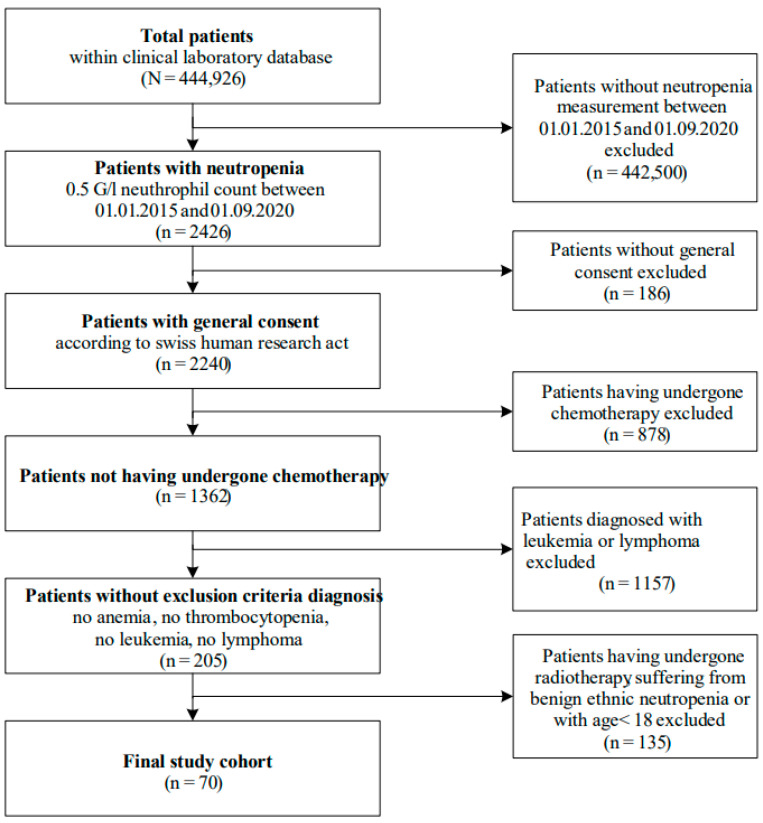
Flowchart of included patients.

**Figure 2 medicina-60-01576-f002:**
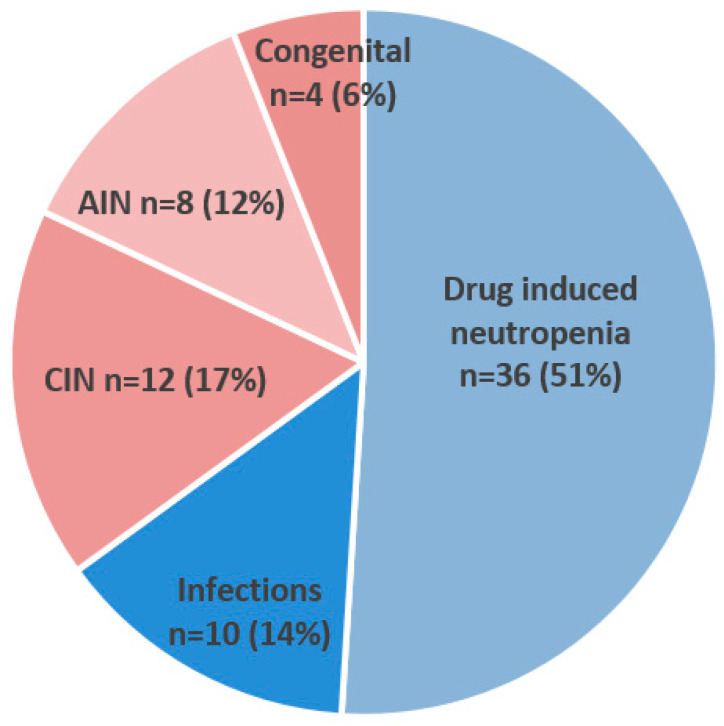
Causes of all isolated severe neutropenia, acute (blue tones) and chronic (rosa tones). Abbreviations: CIN (chronic idiopathic neutropenia), AIN (autoimmune neutropenia).

**Figure 3 medicina-60-01576-f003:**
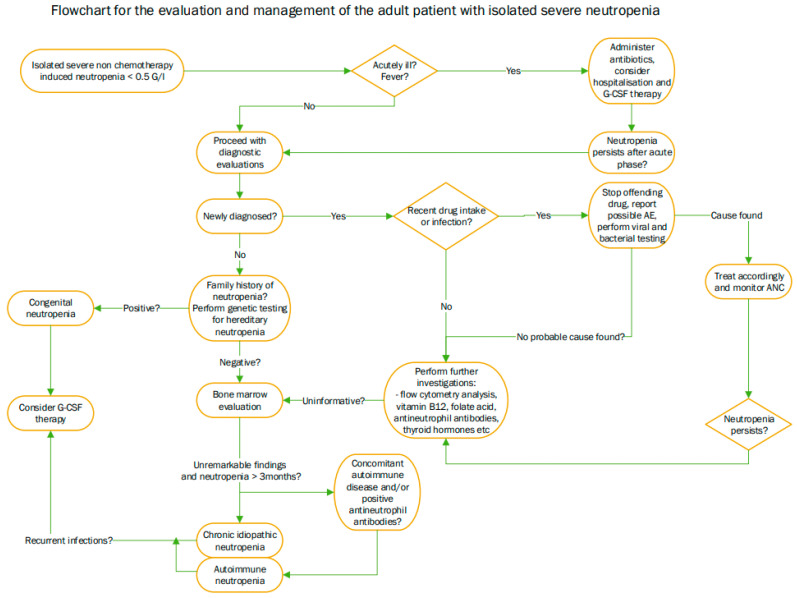
Flowchart for the evaluation and management of severe isolated neutropenia. Abbreviations: AE (adverse event), ANC (absolute neutrophil count), G-CSF (granulocyte stimulating agent).

**Table 1 medicina-60-01576-t001:** Offending drugs and frequencies of drug-induced neutropenia (alphabetical order).

Drug	N	% of Acute Forms
Azathioprin	2	5.5
Clopidogrel	1	2.8
Clozapin	1	2.8
Dimethylfumarat	1	2.8
Infliximab	1	2.8
Lamivudin/zidovudine	2	5.5
Lamotrigin	1	2.8
Metamizole	14	39.0
Methotrexate	2	5.5
Ocrelizumab	1	2.8
Pembrolizumab	1	2.8
Rituximab	4	11.0
Tenofovir/Emtricitabin	1	2.8
Trazodone	2	5.5
Tocilizumab	2	5.5

**Table 2 medicina-60-01576-t002:** Characteristics of post-infectious neutropenia.

Patient	Age at Diagnosis	Pathogen	ANC at Diagnosis (10^9^/L)	Duration of Neutropenia (Days)
1	31	Dengue virus	0.48	1
2	29	*Haemophilus influenzae*	0.22	1
3	39	Enterovirus	0.46	4
4	54	Rhinovirus	0.21	7
5	26	Epstein-Barr virus	0.35	1
6	28	Dengue virus	0.36	2
7	40	Not known	0.42	3
8	48	Not known	0.11	5
9	25	Not known	0.22	1
10	35	Not known	0.42	2

**Table 3 medicina-60-01576-t003:** Frequency of G-CSF use and hospitalization-rates due to infections.

Neutropenia Cause	*n*	G-CSF Use		Hospitalization-Rate, N
		*n*	%	
Congenital neutropenia	4	3	75	3 (75%)
CIN and AIN	20	11	55	2 (10%)
Drug-induced neutropenia	36	8	22	17 (47%)
Post-infectious neutropenia	10	0	0	2 (20%)

Abbreviations: CIN (chronic idiopathic neutropenia), AIN (autoimmune neutropenia).

## Data Availability

The raw data supporting the conclusions of this article will be made available by the authors on request.

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
