# Peer review of "Isolated Severe Neutropenia in Adults, Evaluation of Underlying Causes and Outcomes, Real-World Data Collected over a 5-Year Period in a Tertiary Referral Hospital"

_medicina, 2024, doi:10.3390/medicina60101576_

Round 1
Reviewer 1 Report
Comments and Suggestions for Authors
This manuscript explores the occurrence of isolated severe neutropenia in a tertiary referral hospital using a large population-based cohort study, analyses the suggested diagnostic work-up plan, and identifies the most common underlying causes.
The manuscript itself is well constructed, nicely written. The research presented is essential and has certainly a potential value for clinical readers and raises the clinical relevance of isolated severe neutropenia, however a few potential issues will need to be adequately addressed.
My comments are:
- The authors found a high frequency of metamizole-induced neutropenia in the presented study, however the rate of a potential overdose or overuse of the prescribed drug was not mentioned in detail. Could we assume that all of the recognized cases resulted from severe idiosyncratic adverse drug reactions to metamizole?
- The second most common cause of acute neutropenia in the investigated cohort was due to infection. Were the identified responsible pathogens determined by serological or molecular biology tests? Should the author recommend serum detection or determination of viral copies number during the diagnostic steps?
- The authors described 4 women with chronic neutropenia (3 with CIN and 1 with AIN), who held their pregnancy during the evaluated period. As mentioned, one patient received G-CSF treatment due to recurrent infections, therefore she continued her treatment during pregnancy without complication. What was the recommended dose of the G-CSF treatment, and could you describe the potential use of antibiotics, or extend the recommended antimicrobial or prophylactic treatment?
- The length of G-CSF therapy should be also reported in the detailed paragraph. Have would you monitored the side effects of the G-CSF?
- Please consider increasing the strength of recommendation in the Discussion and highlight the strength of the performed research.
Minor comments to the manuscript:
- On page 2 – line 65. I recommend finishing the sentence “Secondly, we aim to analyze the management and outcome”
- On page 3 – At the figure 1. Please correct the term “benign ethnic neutropenia”.
- On pages 8 – line 263. – Please correct the sentence “We observed that 41% of..”
- I suggest reorganizing the page breakpoints at lines 206, 223,302.
Comments on the Quality of English Language
Minor editing of English language is needed.
Author Response
Comment 1: The authors found a high frequency of metamizole-induced neutropenia in the presented study, however the rate of a potential overdose or overuse of the prescribed drug was not mentioned in detail. Could we assume that all of the recognized cases resulted from severe idiosyncratic adverse drug reactions to metamizole?
Response to comment 1:
We appreciate the reviewer's comment; this is indeed a very important point. We agree that the development of metamizol-induced-neutropenia could be dose-dependent. However, this being a retrospective study, we were not in a position to reconstruct detailed information on the exact dosage taken by every exposed patient. This information (largely self-reporting especially in patients initially treated in other hospitals) was unfortunately not documented in every case. In the cases where metamizol dosage was indeed documented, over-dosage was not observed.
Though over-dosage cannot be excluded with absolute certainty, the data collected strongly suggests that the identified cases resulted from severe idiosyncratic adverse drug reaction.
As this is a valid and important point, we have added the following information on page 10 line 316 of the discussion section:
Though detailed information on the exact dosage and duration of metamizol exposure was not available in this study, it has been shown that dose, in addition to duration of use, may be a risk factor for the development of agranulocytosis associated with metamizol.
(Ref added: Huber, M., et al., Metamizole-induced agranulocytosis revisited: results from the prospective Berlin Case-Control Surveillance Study, Eur J Clin Pharmacol. 2015.
Hedenmalm, K. and O. Spigset, Agranulocytosis and other blood dyscrasias associated with dipyrone (metamizole)), Eur J Clin Pharmacol. 2002)
Comment 2: The second most common cause of acute neutropenia in the investigated cohort was due to infection. Were the identified responsible pathogens determined by serological or molecular biology tests? Should the author recommend serum detection or determination of viral copies number during the diagnostic steps?
Response to comment 2:
The identified responsible pathogens were determined by serological testing.
According to current guidelines (Fioredda F, Papadaki HA. et. al. The European Guidelines on Diagnosis and Management of Neutropenia in Adults and Children: A Consensus Between the European Hematology Association and the EuNet-INNOCHRON COST Action. Hemasphere. 2023 Mar 30;7(4):e872.), patients with isolated acquired neutropenia who have had recent infections should undergo viral and bacterial investigations. Our results are in accordance with this recommendation. Serum testing for viral specific antibodies is generally considered the gold standard for diagnosis, whereas quantification or monitoring of viral load can help inform assessment of risk and strategies for treatment. We therefore recommend antibody studies as the initial diagnostic step.
This paragraph had been added in the discussion, page 11 line 337.
Comment 3: The authors described 4 women with chronic neutropenia (3 with CIN and 1 with AIN), who held their pregnancy during the evaluated period. As mentioned, one patient received G-CSF treatment due to recurrent infections, therefore she continued her treatment during pregnancy without complication. What was the recommended dose of the G-CSF treatment, and could you describe the potential use of antibiotics, or extend the recommended antimicrobial or prophylactic treatment?
Response to comment 3:
The recommended dose of G-CSF in chronic neutropenia patients is 1 to 8 mcg/kg per day, with the effective doses varying by diagnosis: idiopathic: median 1.2 mcg/kg/day; cyclic: median 2.4 mcg/kg/day; congenital: median 7.3 mcg/kg/day (Ref: Dale DC. How I diagnose and treat neutropenia. Curr Opin Hematol. 2016).
This paragraph has been added to page 13 line 410.
The pregnant woman in our cohort who continued this treatment during her pregnancy received a dosage of 3mcg/kg every 2 weeks, this individualized maintenance dose was based on her clinical course as well her ANC. (page 9, line 267)
Many thanks for this vital question on prophylactic antibiotic treatment, which is indeed of great importance in everyday clinical practice. We have added the following paragraph on page 13 line 414.
None of the patients in our cohort with chronic neutropenia received continuous antibiotic prophylaxis. Experts agree on the concept that there is no major advantage in antibiotic prophylaxis in chronic neutropenia rather than to treat the infection at its occurrence.
Comment 4: The length of G-CSF therapy should be also reported in the detailed paragraph. Have would you monitored the side effects of the G-CSF?
Response to comment 4:
Thank you for this important comment. We have added this information in page 12, line 391.
Life-time G-CSF treatment is usually needed in SCN, with individual tailoring of the dose, in order to achieve therapeutic target neutrophil counts ≥1.0 ×109/L. For acquired forms of chronic neutropenia, G-CSF therapy should be recommended for patients with absolute neutrophil count repeatedly less than 0.5 × 109/L with recurrent mouth ulcers, gum disease and or recurrent infections.
(Ref: Dale, D.C. and A.A. Bolyard, An update on the diagnosis and treatment of chronic idiopathic neutropenia.Curr Opin Hematol, 2017. Dale, D.C., How I diagnose and treat neutropenia. Curr Opin Hematol, 2016.)
In this study, we did not study the side-effects of G-CSF therapy in the few patients who received it. As this is undoubtedly a very important point, especially in regards to life-time treatment, we have added the following paragraph on page 13, line 413.
Although G-CSF is safe and efficient in long-term treatment of adult patients with chronic neutropenias possible side effects may include bone pain, arthralgias, myalgias, and headache with onset a few hours after the injections. Less common adverse events include thrombocytopenia, skin rash, injection site reactions, vasculitis, decreased bone density and osteoporosis, though fractures are uncommon (Ref: Heussner P, Haase D, Kanz L, Fonatsch C, Welte K, Freund M. G-CSF in the long-term treatment of cyclic neutropenia and chronic idiopathic neutropenia in adult patients. Int J Hematol. 1995 Dec. Dale DC, Bolyard AA, Schwinzer BG, et al. The severe chronic neutropenia international registry: 10-year follow-up report, Support Cancer Ther. 2006 Jul )
Comment 5: Please consider increasing the strength of recommendation in the Discussion and highlight the strength of the performed research.
Response to comment 5: Many thanks for this suggestion. The paragraph on page 13, line 426 has been edited as follows:
Our study had several strengths. One is the huge number of patients examined, the confirmation of isolated neutropenia with laboratory results, the availability of all laboratory data from additional investigations and the access to clinical data from all of them, making the cohort very consistent. Another strength of the study is the applicability and relevance of the findings to clinical practice, which can lead to more personalized interventions. Another strength of this study is the generalizability due to the large populated investigated. To the best of our knowledge, similar data has not been published to date.
Minor comments to the manuscript:
- On page 2 – line 65. I recommend finishing the sentence “Secondly, we aim to analyze the management and outcome”
Response: This sentence has been corrected.
- On page 3 – At the figure 1. Please correct the term “benign ethnic neutropenia”.
Response: This has been edited in figure 1.
- On pages 8 – line 263. – Please correct the sentence “We observed that 41% of..”
Response: The sentences have been corrected as suggested. Many thanks.
- I suggest reorganizing the page breakpoints at lines 206, 223,302.
Response: We have re-organised and re-worded the paragraphs mentions. We appreciated these comments.
Reviewer 2 Report
Comments and Suggestions for Authors
Comments to the Author:
In their manuscript, Dr. Njue and colleagues conducted a comprehensive retrospective analysis of data obtained through CBC, and diagnostic methods for characterizing neutropenia were bone marrow and cytogenetic and molecular investigations. In addition, blood and other sample cultures and radiological investigations were analyzed to assess infections. This manuscript provides information regarding diagnosing and managing isolated severe neutropenia. They have also subcategorized isolated severe neutropenia based on underlying causes and suggested diagnosing and managing this neutropenic condition. However, I have some minor concerns and suggestions, which are below.
- Do authors use sample size calculation used in descriptive retrospective studies to determine the minimum sample size required for their research, as mentioned by Johnston et al. 2019; BMC Med Res Methodol 2019.? If so, please provide a detailed methodology for this sample size calculation.
- Nutritional deficiency (Folate/ vitamins and copper) is often associated with neutropenia, as reported by Socha et al. Cleve Clin J Med. 2020. Was a nutritional deficiency test performed on these patients at diagnosis? If not, the authors should provide a rationale for not considering it as one of the underlying causes of neutropenia. This would help reconcile whether these patients have nutrition deficiency as one of the underlying causes of neutropenia.
- It would be beneficial for the reader if the authors could provide flow chart diagrams for the diagnosis and management of each subcategory, as they have mentioned in the discussion. These diagrams could be combined into one, providing a clear visual representation of the diagnostic and management processes.
Minor Point
- In Figure 2, the proper color combination should be chosen to make the words inside the diagram visible to the reader. Also, the font size of the legend and abbreviation should be the same.
- The two supplementary tables in lines 115 and 139 support this manuscript but are unavailable for review. The authors should provide these tables for the review.
- There should be a space between the words CIN/ICUS-Nhadrecurrent mentioned in line 263 of the manuscript.
Author Response
We greatly thank the editor and the reviewers for their time, their valuable comments and the opportunity to revise our manuscript: “Isolated Severe Neutropenia in adults, Evaluation of Underlying Causes and Outcomes. Real-world data collected over a 5-year period in a tertiary referral hospital.”
We acknowledge that these recommendations will greatly improve the quality of our work.
Comment 1
- Do authors use sample size calculation used in descriptive retrospective studies to determine the minimum sample size required for their research, as mentioned by Johnston et al. 2019; BMC Med Res Methodol 2019.? If so, please provide a detailed methodology for this sample size calculation.
Response to comment 1: Many thanks for this important question. Due to the retrospective and descriptive nature of this study, we did not use sample size calculation for this study (Kim J, Seo BS. How to calculate sample size and why. Clin Orthop Surg. 2013).
Comment 2
- Nutritional deficiency (Folate/ vitamins and copper) is often associated with neutropenia, as reported by Socha et al. Cleve Clin J Med. 2020. Was a nutritional deficiency test performed on these patients at diagnosis? If not, the authors should provide a rationale for not considering it as one of the underlying causes of neutropenia. This would help reconcile whether these patients have nutrition deficiency as one of the underlying causes of neutropenia.
Response to comment 2
We appreciate the reviewers comment on this very important point. We acknowledge that collecting vital data such as dietary habits of the patients is limited by the retrospective nature of the study as we depended on review of charts that were originally not designed to collect data for research. We retrospectively studied the patients’ electronic charts for any possible cause of neutropenia that was documented. We however did not find any record of nutritional deficiency. We list this as limitation of this study.
Comment 3
- It would be beneficial for the reader if the authors could provide flow chart diagrams for the diagnosis and management of each subcategory, as they have mentioned in the discussion. These diagrams could be combined into one, providing a clear visual representation of the diagnostic and management processes.
Response to comment 3
Thank you for this suggestion which will indeed be helpful for the reader. The flow-chart below has been added to the discussion section of the manuscript (see attached pdf).
Minor Point
- In Figure 2, the proper color combination should be chosen to make the words inside the diagram visible to the reader. Also, the font size of the legend and abbreviation should be the same.
Response: Thank you for this suggestion. The colour as well as the font size have been adjusted accordingly (see attached pdf). - The two supplementary tables in lines 115 and 139 support this manuscript but are unavailable for review. The authors should provide these tables for the review.
Response: We apologise for leaving out the supplementary tables. These have been included for review.
- There should be a space between the words CIN/ICUS-Nhadrecurrent mentioned in line 263 of the manuscript.
Response: This sentence has been corrected. Many thanks.